# Increased Wellbeing following Engagement in a Group Nature-Based Programme: The Green Gym Programme Delivered by the Conservation Volunteers

**DOI:** 10.3390/healthcare10060978

**Published:** 2022-05-25

**Authors:** Nina Smyth, Lisa Thorn, Carly Wood, Dominic Hall, Craig Lister

**Affiliations:** 1School of Social Sciences, Psychology, University of Westminster, London W1W 6UW, UK; l.thorn01@westminster.ac.uk; 2School of Sport, Rehabilitation and Exercise Sciences, University of Essex, Colchester CO4 3SQ, UK; cjwood@essex.ac.uk; 3The Conservation Volunteers, Doncaster DN4 8DB, UK; dominic_hall@hotmail.com; 4Bedfordshire, Luton and Milton Keynes Integrated Care System, Luton LU1 2LJ, UK; craig.lister4@nhs.net

**Keywords:** nature-based activities, nature exposure, conservation, wellbeing, social prescribing

## Abstract

The wellbeing benefits of engaging in a nature-based programme, delivered by the Voluntary, Community and Social Enterprise sector, were examined in this study. Prior to attending The Conservation Volunteers’ Green Gym™, attendees (*n* = 892) completed demographics, health characteristics and the Warwick Edinburgh Mental Wellbeing Short-Form Scale. Attendees (*n* = 253, 28.4%) provided a measure on average 4.5 months later. There were significant increases in wellbeing after engaging in Green Gym, with the greatest increases in those who had the lowest starting levels of wellbeing. Wellbeing increases were sustained on average 8.5 months and 13 months later in those providing a follow up measure (*n* = 92, *n* = 40, respectively). Attendees who continued to engage in Green Gym but not provide follow up data (*n* = 318, 35.7%) tended to be more deprived, female and self-report a health condition. Attendees who did not continue to engage in Green Gym (*n* = 321, 36.0%) tended to be less deprived and younger. These findings provide evidence of the wellbeing benefits of community nature-based activities and social (‘green’) prescribing initiatives and indicate that Green Gym targets some groups most in need.

## 1. Introduction

Exposure to natural environments, such as forests, urban parks, local green spaces and country parks, provide health-related benefits; this is well documented in population, observational, empirical and intervention studies. For example, nature exposure is associated with decreased risk of physical and mental ill-health and disease mortality [1,2,3] and lower use of antidepressants [4]. Nature exposure can also result in reductions in stress [5], and improved wellbeing [6,7,8] and cognition [9]. Exposure to green space is associated with improved physiological function, important for wellbeing and health, such as lower diastolic blood pressure, heart rate and salivary cortisol [10]. Furthermore, living near green space is associated with psychophysiological benefits, such as healthier salivary cortisol profiles [11] and better wellbeing [12].

Over the last decade, there has been growing research demonstrating the health-related benefits of engaging in nature-based activities. Activities that encourage physical activity whilst in nature provide greater physical and health benefits than physical activity or nature exposure alone, and findings seem to be universally obtainable [6,13,14,15,16]. For example, allotment and community gardening are associated with increased physical activity, better wellbeing, mood and general health, as well as reduced social isolation. This holds true for individuals with defined health needs (e.g., people with poor physical and/or mental health and long-term conditions), individuals in care and community settings, and older adults [17,18,19,20]. Nature-based activities may also facilitate improvements in wellbeing [21], which is associated with better health, longer life and better outcomes in chronic conditions, as well as reduced mortality across multiple health conditions [22,23,24,25]. For example, wellbeing increases were observed following engagement in nature-based activities for young people and adults with defined needs (e.g., mental ill-health, vulnerable groups in society). However, the largest increases in wellbeing were observed in individuals with the lowest levels of wellbeing prior to engaging in the nature-based activity, indicating the potential of nature-based interventions for treatment of ill health [26].

Nature-based activities facilitate health-enhancing behaviours, important for wellbeing and health, such as physical activity and social interaction. For example, people who visit local green spaces once per week are four-times more likely to achieve physical activity recommendations compared to those who do not visit [27], and exposure to nature enables individuals to develop social networks, reduce social isolation and promote community belonging [28,29]. In a study by Rogerson et al., participants spent 20% more exercise time outdoors being socially interactive compared to indoors. In this study, social interaction time also significantly predicted intention for future exercise in the outdoors condition but did not in the indoor condition, implying that environments could influence longer-term behavioural choices via a pathway involving social experience [30].

Interacting with natural environments has been shown to have the greatest health-related benefits for individuals from lower socio-economic backgrounds and/or individuals with the lowest levels of wellbeing [5,26,31]. However, a challenge is to ensure those who might most benefit from nature exposure have sufficient opportunities to access quality natural environments, as access to natural environments may be limited. For example, in the UK, 2.69 million people are not within a ten-minute walk of a green space [32] and around a quarter of people visit natural environments, such as green spaces, less than once a month [33]. Moreover, in England, United Kingdom, younger (aged <35 years) and older adults (aged >65 years), individuals from ethnic minorities, those who live in the most deprived areas or adults reporting poorer health and lower life satisfaction have less access to quality natural environments and spend less time outdoors [33,34].

There are a wide range of volunteering programmes that encourage and support communities or individuals to engage in practical activities in natural environments. These activities are typically aimed at making improvements or conserving local spaces, such as creating and restoring natural habitats, the maintenance of amenities in outdoor spaces, such as parks, footpaths, trails and building physical features in the natural environment. Qualitative studies with individuals volunteering in nature-based activities report a wide range of health-related benefits, such as increased fitness, general health, quality of life and wellbeing [35,36,37,38,39,40]; however, reports are mostly based on individuals from white backgrounds, higher socio-economic groups and/or those who report better health and wellbeing. Evidence from quantitative studies, in which the health-related changes from engagement in the nature-based programme are examined, show mixed results. For example, studies used different study designs, such as changes pre and post volunteering in conservation activities or participants allocated to an intervention or control group; some studies reported increases in mental and emotional wellbeing and others did not observe any changes [41]. More research is needed to determine the health-related benefits in more diverse samples with equal representation of genders and ethnicity.

In health care, targeted nature-based activities are prescribed alongside usual treatment for individuals with defined needs, such as mental health issues and noncommunicable diseases; these are known as ‘green social prescriptions’ [42]. Nature-based social prescribing, such as gardening, conservation activities and ecotherapy (contact with nature in a facilitated, structured and safe way), have been shown to improve social connectedness, strengthen social networks, reduce stress and improve health and wellbeing.

Green social prescribing is typically delivered through the Voluntary, Community and Social Enterprise (VCSE) sector [43]. There is evidence of the cost effectiveness for such programmes, for example, a social return on investment has been demonstrated [44,45]. However, the health-related benefits of these programmes in the VCSE sector, such as increased social cohesion, increases in physical activity and better wellbeing (see [45,46]), are limited to small-scale evaluations of specific programmes and simple reporting of descriptive statistics or reports derived from qualitative interviews. With the increased popularity in community-based or the social prescribing of nature-based activities, there is increased demand for services from the VCSE sector and, thus, a strain on funding and resources [47]. Financial and practical support are needed to ensure the effective delivery of activities and access for people from different socio-economic and ethnic groups [48,49]. Thus, evidence for the effectiveness of these types of programmes is essential. Robust evidence-based research, capturing routine assessment of programme efficacy in promoting health and wellbeing, is needed for the full benefits of volunteering programmes and green social prescribing to be seen [50,51,52].

The primary aim of this study was, therefore, to examine the wellbeing benefits of a voluntary nature-based programme.

## 2. Materials and Methods

### 2.1. Participants

Participants (*n* = 892) were individuals engaging in a nature-based programme delivered by ‘The Conservation Volunteers’ (TCV) Green Gym between June 2017 and March 2020. Participants included 440 (49.3%) males and 439 (49.2%) females (13 individuals did not report their gender) and were aged on average 48.90 (*SD* = 17.6) years old (6 individuals did not report their age). Participants included those who self-selected to engage in the programme and those referred to the programme via social prescribing and/or other health and social-care routes. Participant referral pathway was not identified.

### 2.2. The Nature-Based Programme

The TCV Green Gym programme is a national programme with over 60 gyms located throughout England, Scotland and Northern Ireland. Each Green Gym has on average between 50 and 60 Green Gym attendees. Green Gym is based on social action theory and the Five Ways to Wellbeing initiative, which provides the opportunity for individuals, regardless of social background, to connect with other people and nature, be physically active, learn new skills, give to others and be mindful [53]. Attendees of Green Gym complete a range of practical activities, such as planting trees, managing wildflowers, community growing or path improvements, with nature connection activities, such as learning about species or habitats, designed to improve local green spaces and for attendees to acquire new skills, knowledge and confidence. The group-based activity offers opportunity for peer support and thus opportunity to socialise with others and to develop social networks. Peer support is delivered in many different ways; initially, just turning up to an unknown creates a relationship or group of peers, all having in common that they have made the decision to join Green Gym. The beginning of each programme is a group discussion on the tasks at hand and warm up, allowing attendees to share in the decision making of who will do what, creating a peer-to-peer environment of mutual support and learning. Within and outside of sessions it can be common for more confident or able participants to provide support to other attendees. In some cases, more formal ‘buddying’ may be set up, for example to collect people to walk to a site. Many of the Green Gym leaders were once attendees of Green Gym; this allows them to share their journey as a peer, rather than an expert or medical professional that participants may have previously been used to, breaking down professional barriers. Handing out of Green Gym t-shirts creates a group environment, visually establishing one’s peers within the open space that is being managed. Green Gym is delivered in a range of natural environments, such as urban natural spaces and woodlands, on a weekly basis, usually during the daytime. Whilst regular participation in Green Gym is encouraged, regardless of how attendees join Green Gym, they can attend sessions on a flexible basis and are not committed to attend sessions consecutively, for a set number of sessions or for a set period. Typically, between 6 and 20 individuals participate in any single Green Gym session, which lasts between three and four hours. Sessions are led by a trained Green Gym Project Officer or Volunteer Co-ordinator, starting with a warm up and ending with a cool down.

### 2.3. Procedure

TCV asked Green Gym attendees to complete a paper survey prior to and after engagement in Green Gym sessions. TCV assigned a unique ID number for each participant which was assigned to each survey they completed. The exact number of sessions participants engaged in Green Gym were unknown, but attendees can attend Green Gym sessions on a weekly basis. All data collection and anonymisation of responses were completed by TCV. Completion of the survey was voluntary with Green Gym attendees giving written informed consent prior to completing the survey. Attendees were made aware that their data would be handled in accordance with the General Data Protection Regulations (2018) and the Data Protection Act (1988) depending on when the survey was completed. They were also informed that their data would be shared with a third party in an anonymised form. Ethical approval for transfer of data and analysis of data was obtained from the University of Westminster.

### 2.4. Measures

Green Gym attendees were asked to provide demographic information, such as age and sex and their postcode in order to determine their index of multiple deprivation decile (IMD). IMD provides a measure of deprivation according to the area individuals live in [54]. Volunteer deprivation decile IMD score was categorised as quintiles with the bottom quintile representing individuals living in the most deprived areas and the top quintile representing individuals living in the least deprived areas of the UK. Attendees were also asked to report if they had any known physical and/or mental health conditions and if they had difficulty performing activities. Responses were dichotomised to create two health status variables: health condition reported and no health condition reported; two activity status variables were also created: difficulty performing activities reported and no difficulty performing activities reported.

Mental wellbeing was assessed by the Warwick Edinburgh Mental Wellbeing Short-Form Scale (SWEMWBS) [55], a global measure of mental wellbeing comprising affective–emotional aspects, cognitive–evaluative dimensions and psychological functioning, used to monitor changes in wellbeing. The SWEMWBS consists of seven positively worded items from the full 14-item scale [56] and is scored by summing responses to each item, which are scored on a five-point Likert scale from 1 (none of the time) to 5 (all of the time). Scores range between 7 and 35, with higher scores indicating better wellbeing. The SWEMWBS has been validated for use in the UK general population and shows good internal consistency (∝ = 0.84) [57] with high correlations between the full and short version r = 0.954 [55]. In the current sample the Cronbach’s alpha was 0.87, indicating very good reliability. In line with recommendations [55] the total raw scores were transformed into metric scores using the SWEMWBS conversion table. Metric-converted SWEMWBS scores were categorised in relation to UK population mean and standard deviation (SD) values. Scores below one SD of the mean were categorised as ‘low’ wellbeing, scores within one SD of the sample mean were categorised as ‘average’ wellbeing and scores above one SD of the mean were categorised as ‘high’ wellbeing [57]. For low SWEMWBS, scores ranged between 9.00 and 19.25, for ‘Average’ wellbeing, scores ranged between 19.98 and 27.03 and for ‘High’ wellbeing, scores ranged between 28.13 and 35.00. Wellbeing categories were dichotomised to create categories of ‘low wellbeing’ and ‘average to high wellbeing’.

### 2.5. Treatment of Data and Statistical Analysis

Analyses were conducted using SPSS 25 (IMB) and were initially conducted on baseline data from all participants (*n* = 892). Skewness and kurtosis scores indicate the wellbeing (SWEMWBS) data was normally distributed. Baseline wellbeing (SWEMWBS) scores were examined in relation to demographic and health/activity status variables. Correlations with age and deprivation quintile were conducted and between-subjects ANOVAs were used to examine differences in sex, self-reported health/activity status and Green Gym engagement status, with the latter reflecting attendees who continued Green Gym versus those that did not, and attendees who completed the first follow up measure versus those who did not. To examine if there were differences or relationships in demographic and health/activity status with Green Gym engagement, chi-square tests were conducted to examine associations with gender and health/activity status variables and one-way ANOVAs were conducted to examine differences in age and IMD deprivation quintile.

To examine the effect of engagement in Green Gym over time, analyses were conducted on participants who provided a follow up measure (*n* = 252). Multi-level modelling was used to investigate changes in baseline SWEMWBS scores following engagement in Green Gym. Comparisons were made between baseline and the first follow up. In the model, participant identity was the subject variable and survey-time point (baseline versus follow up 1) was modelled as repeated effects. A compound symmetry structure was adopted in the model. Attendees completed the first follow up measure at different times, thus, the number of days between completing the baseline and follow up measure was entered as a covariate into the model. The demographic and health/activity status variables that were associated with wellbeing scores at baseline were examined in relation to Green Gym associated changes in SWEMWBS scores. Age, IMD deprivation quintile, health/activity status and ‘low’ and ‘average to high’ wellbeing category were entered separately into the model to examine if effects remained after controlling for these variables and to enable examination of potential interacting effects. Subsequent analyses were conducted to examine if changes in SWEMWBS remained at the second and the third follow up points; the mixed regression model included SWEMWBS at baseline and follow up 1, 2 and 3 and AR(1) solutions were modelled. One-sample t-tests were conducted comparing SWEMWBS scores at baseline and follow up points 1–3 in relation to UK normative values [57].

## 3. Results

Participants’ mean index of multiple deprivation decile (IMD) was 4.5 (*SD* = 2.8, 32 individuals did not provide IMD). Participants included individuals who self-reported a health condition (*n* = 367, 41.1%) and individuals who reported having difficulties doing activities (*n* = 169, 18.9%, one person did not report their health or activity status). Green Gym attendees’ baseline SWEMWBS score was significantly positively correlated with age, *r* = 0.12, *p* < 0.001 and IMD deprivation quintile, *r* = 0.16, *p* < 0.001. Baseline SWEMWBS score did not differ between males and females, *F* = 0.15, *p* = 0.695. Baseline SWEMWBS score was significantly lower for Green Gym attendee’s reporting a health condition (*M* = 21.5, *SD* = 4.4) compared with those who did not report a health condition (*M* = 23.6, *SD* = 4.3), *F* = 50.78, *p* < 0.001. Baseline SWEMWBS score also significantly differed for attendees who reported difficulties performing activities (*M* = 23.1, *SD* = 4.4) compared with those who did not report any difficulties performing activities (*M* = 21.5, *SD* = 4.8), *F* = 16.63, *p* < 0.001. A one-sample t-test revealed that baseline SWEMWBS score for Green Gym attendees was significantly lower than UK population norm score, *t* = 6.14, *df* = 882, *p* < 0.001, see Figure 1.

After providing a baseline measure, 28.4% (*n* = 253) of Green Gym attendees provided a follow up survey. However, 35.7% (*n* = 318) of Green Gym attendees continued to engage in Green Gym but did not provide a follow up measure, whilst 36.0% (*n =* 321) attendees did not continue to engage in Green Gym after completing the baseline measure. Baseline SWEMWBS score was not significantly different between these groups, *F*_(2,880)_ = 0.34, *p* = 0.711. Deprivation quintile differed between the three groups, *F*_(1, 857)_ = 7.38, *p* < 0.001); those who were the least deprived tended to leave Green Gym, whilst those who were the most deprived tended to stay but did not provide a follow up measure. Age was also significantly different between the three groups, *F*_(1, 883)_ = 28.01, *p* < 0.001; attendees who left Green Gym were significantly younger and those that provided follow up data were significantly older. A similar number of males (*n* = 153, 48.9%) and females (*n* = 160, 51.1%) left Green Gym and those who continued to engage in Green Gym but did not provide a follow up measure were more likely to be female (*n* = 170, 54.1%), whilst those who provided follow up data were more likely to be male (*n* = 143, 56.7%), *𝛘^2^* (2) = 6.896, *p* = 0.032. The proportion of participants with a health condition was greater in the group that continued engaging in Green Gym without providing a follow up measure (*n* = 162, 50.9%), in comparison to those that continued engaging in Green Gym and provided a follow up measure (*n =* 84, 33.2%) and those that left Green Gym (*n* = 121, 37.8%), *𝛘^2^ (*2) = 20.660 *p* < 0.001. The wellbeing category or difficulty performing activities did not influence engagement with Green Gym or whether a follow up measure was provided, *p* > 0.05.

To examine the effect of engaging in Green Gym on SWEMWBS scores, analyses were conducted on Green Gym attendees providing a baseline measure and a follow up measure. Multi-level modelling was conducted to examine changes in SWEMWBS scores prior to engaging in Green Gym and at the first follow up point, completed on average 4.5 months later. There was a significant increase in SWEMWBS scores between baseline and at follow up, *F* = 18.67, *df =* 1, 248.61, *p* < 0.001. To examine the potential interactive effects of other measured variables on the main finding, age, health condition status and difficulties performing activities, the deprivation quintile and wellbeing category were entered separately into the model. Increases in SWEMWBS scores were more pronounced for participants who had low SWEMWBS scores, *F* = 18.78, *df =* 1, 246.14, *p* < 0.001, see Figure 2. In all instances, the increase in SWEMWBS scores from baseline to the first follow up remained significant and no other variable interacted with changes in SWEMWBS score.

A sub-group of attendees completed further follow up measures. A second follow-up measure was completed by attendees on average 8.5 months later (*n* = 92) and a third follow up measure on average 13 months later (*n* = 40), following the baseline measure. These follow up points were included in separate multi-level models to tentatively examine if increases in SWEMWBS scores remained. Increases in SWEMWBS scores were associated with time point, *F* = 8.54, *df* = 3, 416.68, *p* < 0.001. Increases in SWEMWBS scores between baseline and the first follow up remained *p* < 0.001 in this reduced sample. Moreover, there was a significant increase in SWEMWBS scores between baseline and follow-up 3, *p* = 0.011. There were no other significant differences in SWEMWBS scores between the other time points, *p* > 0.05. At all follow up points, there were no significant differences in the mean SWEMWBS score between the Green Gym attendees and UK normative values, see Figure 1.

## 4. Discussion

The aim of the study was to examine the wellbeing benefits of a group nature-based activity, TCV’s Green Gym. Green Gym which is delivered in the community, and designed to encourage physical activity and social interaction whilst in nature. The results of the current study demonstrated overall improvements in wellbeing, following engagement in Green Gym, on average, 4.5 months later. Prior to engaging in Green Gym, levels of wellbeing were lower for Green Gym attendees compared with UK normative values. However, following engagement in Green Gym, there were no differences in wellbeing levels. These findings add to the emerging body of evidence showing that engaging in group nature-based activities is associated with wellbeing benefits for adults [13,15,18,19,26]. Moreover, increases in wellbeing were more pronounced for Green Gym attendees who were categorized as having ‘low’ levels of wellbeing prior to engaging in Green Gym; this finding is consistent with studies showing that engagement in nature-based activities, such as horticultural and green exercise programmes, is most beneficial for individuals presenting with low levels of wellbeing compared to individuals presenting with average and high levels of wellbeing [18,26]. A sub-sample of attendees who continued to engage in Green Gym also provided two further follow up measures. Analyses revealed improvements in wellbeing were sustained, on average, 8.5 months after and further increases in wellbeing were observed, on average, 13 months later.

Green Gym is delivered by TCV, which is part of the VCSE sector and is an example of a voluntary community nature-based programme. The findings from the current study add to the evidence demonstrating the health-related benefits of nature-based volunteering programmes [35,36,37,38,39,40,58,59]. Nature-based activities are being used in health and social care [50,60] and feature in the UK Government’s green social prescribing initiative (NHS England) to help tackle the increasing mental health problems and poor wellbeing associated with the COVID-19 pandemic. Green Gym attendees included individuals who were referred via health and social care or social prescribing routes. Thus, the current findings add to the evidence base showing the health-related benefits of nature prescriptions [42] and green social prescribing [61]. Evidence on the efficacy of nature-based programmes as prescriptions is needed, given the increased demand for social prescribing ‘green’ initiatives [47].

It is unsurprising that a third of Green Gym attendees did not provide a follow up measure; this is a difficulty with gathering evidence-based research on the benefits of community or social prescribing programmes [50,51]. Although wellbeing levels for this group were not significantly different to Green Gym attendees who continued to engage with Green Gym and provide follow up data, the lack of follow up data may have resulted in the benefits of Green Gym being underestimated. For example, Green Gym attendees who continued to engage in Green Gym but not provide follow up data tended to be more deprived. Previous research indicates that exposure to nature most benefits individuals from lower socio-economic backgrounds, e.g., [18,31]. A third of Green Gym attendees also left Green Gym before providing a follow up measure. The reasons for leaving Green Gym are unknown. However, we found that attendees who left tended to be less deprived, which indicates that Green Gym may be targeting those individuals who are most deprived. However, those that left Green Gym also tended to be younger whilst those who continued to engage in Green Gym tended to be older. In line with previous research showing increases in wellbeing in real-world nature-based programmes completed over several weeks/months [26], we did not find that age interacted with increases in wellbeing; however, engagement in shorter bouts of green exercise have previously been shown to be associated with more-pronounced health-related benefits for older adults [13,62]. Sex and health status did not influence whether an individual decided to continue to engage with Green Gym; however, being female and reporting a health condition were associated with not providing follow up data. A challenge for the evaluation of real-world community-led nature-based programmes is to involve attendees in the evaluation programme, in order to examine the health-related benefits. To increase future evaluation response rates, it may be useful for the VCSE sector to involve attendees of Green Gym in the design, collection and dissemination of evaluation data to obtain better-quality data for robust evaluation on the efficacy of Green Gym.

Previous nature-based evaluations are limited to reporting simple descriptive statistics or reports from qualitative interviews, and these are based on small samples or individual programmes [49,50,51]. The current study is the first evaluation, using statistical methods, to examine the wellbeing benefits following engagement in Green Gym, across several Green Gyms throughout the UK, over multiple time points. This is the first systematic evaluation of the TCV Green Gym with a larger cohort of Green Gym attendees included. Baseline data represented approximately 30% of Green Gym attendees and the first follow up data represent approximately 10% of Green Gym attendees. The current sample of Green Gym attendees included individuals from a wide age range, with good representation across all age categories and equal representation of males and females, whilst previous evaluations have seen lower representation from younger age groups (i.e., ages below 54 years) and under-representation of females. Compared with previous small-scale TCV Green Gym evaluations, the current sample included a higher proportion of Green Gym attendees who report a health condition. Level of deprivation was obtained for the current sample and there was a good spread of individuals across all deprivation categories, with a slightly higher percentage of individuals from more deprived areas.

Multiple measurements of wellbeing over time made it possible to examine if improvements in wellbeing were sustained over the longer-term. Previous evaluations of nature-based activities have examined the changes in wellbeing over shorter periods, for example, three to six months [26]. However, in the current study, these sustained improvements were tentatively examined, since the sample was reduced to a third of the sample for the second follow up and 15% of the sample for the third follow up. Whilst attendees were asked to complete the baseline survey on the first Green Gym session, it is possible that some attendees completed it after their first session, thus, diluting the potential wellbeing benefits evidenced. Days between completing the baseline measure and follow up measures varied; thus, it was not possible to conduct analyses on a set number of days/weeks (e.g., 3, 6 and 12 months). Due to the flexible basis of attendees engaging in Green Gym, it was not possible to determine how regularly participants engaged in Green Gym at each time point and determine whether frequency of engagement is related with better wellbeing. Green Gym attendees either self-selected to engage in Green Gym or were referred via health or social prescribing routes; it was not possible to determine referral pathways for Green Gym attendees. Further research could examine the efficacy of Green Gym for different referral pathways. Ethnicity data were not provided for this sample. Previous evaluations of Green Gym have seen under representation of non-White individuals [46]. Future research should examine whether health-related benefits are present in ethnic minority groups. Up to a third of participants did not continue to engage in Green Gym, and it would be useful to explore reasons for continuing and discontinuing engagement with Green Gym. Moreover, younger participants were more likely to leave Green Gym; it may, therefore, be beneficial for programmes to be tailored for young adults and teenagers to increase retention [63]. Thus, future research on nature-based programmes delivered via the VCSE sector should understand how programmes address the needs of diverse groups.

## 5. Conclusions

Green Gym is an example of a voluntary nature-based programme or social prescription. The programme is designed to engage attendees in group-based physical activity with a purpose, in nature settings. The findings suggest that Green Gym can facilitate wellbeing improvements. Increases in wellbeing were observed, on average, 4.5 months after engaging in Green Gym and were sustained, on average, 8.5 months later, and further increases were observed, on average, 13 months later. Increases in wellbeing were more pronounced for individuals with low levels of wellbeing.

## Figures and Tables

**Figure 1 healthcare-10-00978-f001:**
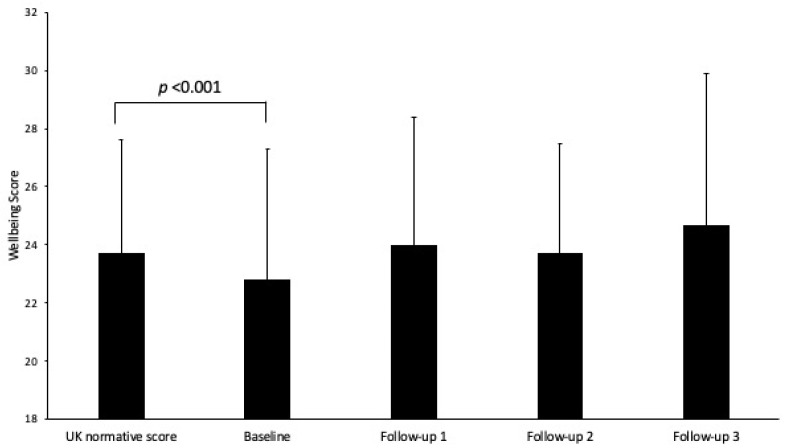
Mean (±SD) SWEMWBS normative scores and Green Gym attendee scores at baseline and follow up points.

**Figure 2 healthcare-10-00978-f002:**
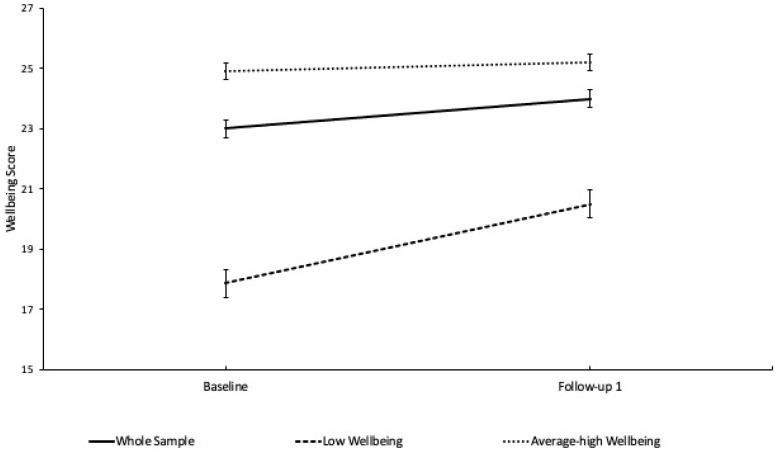
Mean (±SEM) SWEMWBS scores at baseline and follow up (on average 4.5 months later).

## Data Availability

Data will be made available on request.

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
