# Peer review of "Increased Wellbeing following Engagement in a Group Nature-Based Programme: The Green Gym Programme Delivered by the Conservation Volunteers"

_healthcare, 2022, doi:10.3390/healthcare10060978_

Round 1

Reviewer 1 Report

My congratulations this work is well constructed. The methodology is clear, the results are well explained. The conclusions are present.  The authors present limitations in this research and have considered alternative explanations of the results.  The paper is very clear, and has the link between the objective of your study, approach, and results. I haven't a suggestion to correct.

Author Response

Thank you for your positive comments about the article.

Reviewer 2 Report

This study presented the effects of green gym programs in UK and its contents and results are brilliant!

However, please add below matters.

  1. Please add the explanation about nature based program (page 3 133- 135).  How do attendees receive peer support and what kind of peer supports are needed by attendee?
  2. There have been many papers and studies concerned with occupational activities which are similar with green gym.    

Author Response

This study presented the effects of green gym programs in UK and its contents and results are brilliant!

Thank you for your positive comments about the article and your suggestions below.

However, please add below matters.

  1. Please add the explanation about nature based program (page 3 133- 135).  How do attendees receive peer support and what kind of peer supports are needed by attendee?

We have added the following:

Peer support is delivered in many different aspects; initially just turning up to an unknown creates a relationship or group of peers, all having in common that they have made the decision to join green gym. The beginning of each programme is a group discussion on the tasks at hand and warm-up, allowing attendees to share in the decision making of who will do what, creating a peer-to-peer environment of mutual support and learning. Within and outside of sessions it can be common for more confident or able participants to provide support to other attendees. In some cases, more formal 'buddying' may be set up, for example to collect people to walk to a site. Many of the green gym leaders were once attendees of green gym, this allows them to share their journey as a peer, rather than an expert or medical professional that participants may have previously been used to, breaking down professional barriers. Handing out of green gym T-shirts creates a group environment, visually establishing one’s peers within the open space that is being managed (see lines 136-148, pg 3).

  1. There have been many papers and studies concerned with occupational activities which are similar with green gym.    

Green gym is focused on conservation activities in groups in the local natural environments. Whereas nature-based strategies in occupational settings are focused more on walking interventions or changing workplace environments, thus we feel that this would take away the focus of the introduction. However, if there are specific studies that you would like to suggest we would welcome the references to these.

Reviewer 3 Report

The article entitled ,,Increased wellbeing following engagement in a group nature-based programme: The Conservations Volunteers green gym programme” is interesting and could have an impact on the science community. The authors have been shown the first evaluation, using statistical methods, to examine the wellbeing benefits following engaging in green gym, across several green gyms throughout the UK, over multiple time points. This is the first systematic evaluation of the TCV green gym with a larger cohort of green gym attendees included. The cited references are current: 36 out of 63 within the last 5 years.

My remarks and recommendations are as follows:

  1. Page 4, line 187: The name of the used statistical programme is missing (e.g. Excel, SPSS version….).
  2. Page 4, line 189: ANOVA test be considered in case of SWEMWBS scores’ variable, should be mentioned its distribution, and paste the box-plots about it (page 5, line 232).

If contains outlier data or the distribution is not normally, please use the proper non-parametric tests. If the box-plot represents normality and use ANOVA, please paste the 95% of confidence interval each case.

  1. From page 5, line 223: Please give the statistical values as follows: r=0.15; p<0.001 and so on (with zero before the decimal point).

It can be recommended for publication in Special Issue of Outdoor and Nature Therapy for Healthcare, however after minor revision.

Author Response

The article entitled ,,Increased wellbeing following engagement in a group nature-based programme: The Conservations Volunteers green gym programme” is interesting and could have an impact on the science community. The authors have been shown the first evaluation, using statistical methods, to examine the wellbeing benefits following engaging in green gym, across several green gyms throughout the UK, over multiple time points. This is the first systematic evaluation of the TCV green gym with a larger cohort of green gym attendees included. The cited references are current: 36 out of 63 within the last 5 years.

Thank you for your positive comments about the article and your suggestions below.

My remarks and recommendations are as follows:

  1. Page 4, line 187: The name of the used statistical programme is missing (e.g. Excel, SPSS version….).

We have added this in.

Analyses were conducted using SPSS 25 (IMB) and were initially conducted on baseline data from all participants (N = 892) (see line 203, pg 5).

  1. Page 4, line 189: ANOVA test be considered in case of SWEMWBS scores’ variable, should be mentioned its distribution, and paste the box-plots about it (page 5, line 232).

If contains outlier data or the distribution is not normally, please use the proper non-parametric tests. If the box-plot represents normality and use ANOVA, please paste the 95% of confidence interval each case.

We are unclear what this point is about. On pg 4, line 189, we are referring to correlations or the Cronbach’s alpha score. We did not use ANOVA to examine differences in SWEMWBS scores between timepoints – we used mixed regression modelling. However, we do insert a sentence on pg 5 (line 204-205) that Skewness and kurtosis scores indicate the wellbeing (SWEMWBS) data was normally distributed’.

  1. From page 5, line 223: Please give the statistical values as follows: r=0.15; p<0.001 and so on (with zero before the decimal point).

APA guidelines state that a leading 0 should not be used for numbers that cannot exceed  1 (e.g. for r and p values). We provide the zero before the decimal point for values that can exceed 1 (e.g. F values).

It can be recommended for publication in Special Issue of Outdoor and Nature Therapy for Healthcare, however after minor revision.

Reviewer 4 Report

Dear Authors,

the survey is interesting and valuable.  Yours findings help to expand the knowledge on the matter. The Introduction is well-written and has suitable references. I've made some comments to make the text clearer and more understandable for readers. 

Line 114 - 2. Materials and Methods 2.1. Participants - Questionnaires were delivered for a long term - from June 2017 to March 2020. You write that there were 892 participants. How do you know that, some participants didn’t fulfill the paper survey twice or more?  In my opinion, the description of the procedure of the survey should be improved. 

Line 143 - 2.3. Procedure  - Green gym attendees were asked to complete a paper survey prior to and after engagement in green gym sessions. The exact number of sessions were unknown, but attendees can attend green gym sessions on a weekly basis. -  If the exact number of sessions were unknown, how can we compare results between one who attended green gym e.g. every week with one who was there from time to time, or e.g. only once? I mean, isn’t it a big simplification to lump all participants together and treat them in the same way?

Line 204 -206 - Why did you decide to do the first follow up measure at different times (number of days)? Wouldn’t it be more reliable to make between baseline and the first follow-up (and next ones) in the strictly determined number of days? Have you got any references for your decision?

Best regards!

Author Response

the survey is interesting and valuable.  Yours findings help to expand the knowledge on the matter. The Introduction is well-written and has suitable references. I've made some comments to make the text clearer and more understandable for readers. 

Thank you for your positive comments about the article and for the suggestions below.

Line 114 - 2. Materials and Methods 2.1. Participants - Questionnaires were delivered for a long term - from June 2017 to March 2020. You write that there were 892 participants. How do you know that, some participants didn’t fulfill the paper survey twice or more?  In my opinion, the description of the procedure of the survey should be improved. 

At baseline and the three follow up points, participants completed the survey once. This is identified by the unique ID number TCV assigned to each participant survey response. We have made it clear in section 2.1 that TCV assigned a unique number to participants survey responses:

TCV asked green gym attendees to complete a paper survey prior to and after engagement in green gym sessions. TCV assigned a unique ID number for each participant which was assigned to each survey participants completed (see lines 157-160, pg 4).

Line 143 - 2.3. Procedure  - Green gym attendees were asked to complete a paper survey prior to and after engagement in green gym sessions. The exact number of sessions were unknown, but attendees can attend green gym sessions on a weekly basis. -  If the exact number of sessions were unknown, how can we compare results between one who attended green gym e.g. every week with one who was there from time to time, or e.g. only once? I mean, isn’t it a big simplification to lump all participants together and treat them in the same way?

One of the advantages of TCV green gym is that participants can engage in green gym on a flexible basis, thus they do not need to commit to a set number of sessions over a set period of time. However, this makes it difficult for TCV to determine the number of sessions participants completed. We have acknowledged this as a limitation in the discussion:

Due to the flexible basis of attendees engaging in green gym, it was not possible to determine how regularly participants engaged in green gym at each time points and determine whether frequency of engagement is related with better well-being (see lines 384-387, pg 10).

We also made this clearer in the procedure section:

The exact number of sessions participants engaged in green gym were unknown, but attendees can attend green gym sessions on a weekly basis (see 158-160, pg 4).

Line 204 -206 - Why did you decide to do the first follow up measure at different times (number of days)? Wouldn’t it be more reliable to make between baseline and the first follow-up (and next ones) in the strictly determined number of days? Have you got any references for your decision?

The number of days between baseline and follow up points vary. Thus, it was not possible to conduct analyses on a determined number of days (e.g. 3, 6 and 12 months). TCV collected the data as stated in the methods thus we had no control of when to ask green gym attendees to complete a follow up measure. We have acknowledged this as a limitation:

Days between completing the baseline measure and a follow up measures varied; thus it was not possible to conduct analyses on a set number of days/weeks (e.g. 3, 6 and 12 months) (see lines, 382-387, pg 10).

Reviewer 5 Report

Congratulations on an important, interesting and impactful study. The research is new as it is the first evaluation, using quantitative statistical methods, to investigate the wellbeing benefits which flow from engagement in green gym. In addition the data is collected across several green gyms throughout the UK, over time points in time. Just a couple of minor suggestions:

The second part of the title could be refined for further clarity. Perhaps: The green gym programme for conservation volunteers.

Future research could be recommended particularly examining why people leave the "green gym"

Author Response

Congratulations on an important, interesting and impactful study. The research is new as it is the first evaluation, using quantitative statistical methods, to investigate the wellbeing benefits which flow from engagement in green gym. In addition the data is collected across several green gyms throughout the UK, over time points in time. Just a couple of minor suggestions:

Thank you for your positive comments about the article.

The second part of the title could be refined for further clarity. Perhaps: The green gym programme for conservation volunteers.

Thank you for this recommendation, we have made the following revision to the title:

Increased wellbeing following engagement in a group nature-based programme: The green gym programme delivered by the Conservations Volunteers.

Future research could be recommended particularly examining why people leave the "green gym"

Thank you for this suggestion, we have made this addition:

Up to a third of participants did not continue to engage in green gym, it would be useful to explore reasons for continuing and discontinuing engagement with green gym (see lines 393-395, pg 10).